# Dual-format attentional template during preparation in human visual cortex

**Yilin Chen[1], Taosheng Liu[2], Ke Jia[3,4,5]\*, Jan Theeuwes[6], Mengyuan Gong[1]\***

[1]Department of Psychology and Behavioral Sciences, Zhejiang University, Hangzhou, China; [2]Department of Psychology, Michigan State University, East Lansing, United States; [3]Liangzhu Laboratory, MOE Frontier Science Center for Brain Science and Brain-machine Integration, State Key Laboratory of Brain-machine Intelligence, Zhejiang University, Hangzhou, China; [4]Department of Neurobiology, Affiliated Mental Health Center & Hangzhou Seventh People's Hospital, Zhejiang University School of Medicine, Hangzhou, China; [5]NHC and CAMS Key Laboratory of Medical Neurobiology, Zhejiang University, Hangzhou, China; [6]Department of Experimental and Applied Psychology, Vrije Universiteit Amsterdam, Amsterdam, Netherlands

**\*For correspondence:**
kjia@zju.edu.cn (KJ);
gongmy426@zju.edu.cn (MG)

**Competing interest:** The authors declare that no competing interests exist.

## eLife Assessment

By combining the 'pinging' technique with fMRI-based multivariate pattern analysis, this **important** study provides **compelling** evidence for a dual-format representation of attention during the preparatory period. The findings help reconcile the debate between sensory-like and non-sensory accounts of attentional templates and shed light on how the brain flexibly deploys different forms of templates to guide attention. This work will be of broad interest to researchers in psychology, vision science, and cognitive neuroscience.

**Abstract** Goal-directed attention relies on forming internal templates of key information relevant for guiding behavior, particularly when preparing for upcoming sensory inputs. However, evidence on how these attentional templates are represented during preparation remains controversial. Here, we combine functional magnetic resonance imaging with an orientation cueing task to isolate preparatory activity from stimulus-evoked responses. Using multivariate pattern analysis, we found decodable information about the to-be-attended orientation during preparation; yet preparatory activity patterns were different from those evoked when actual orientations were perceived. When perturbing the neural activity by means of a visual impulse ('pinging' technique), the preparatory activity patterns in visual cortex resembled those associated with perceiving these orientations. The observed differential patterns with and without the impulse perturbation suggest a predominantly non-sensory format and a latent, sensory-like format of representation during preparation. Furthermore, the emergence of the sensory-like template coincided with enhanced information connectivity between V1 and frontoparietal areas and was associated with improved behavioral performance. By engaging this dual-format mechanism during preparation, the brain is able to encode both abstract, non-sensory information and more detailed, sensory information, potentially providing advantages for adaptive attentional control. For example, consistent with recent theories of visual search, a predominantly non-sensory template can support the initial guidance and a latent sensory-like format can support prospective stimulus processing.

## Introduction

To address the challenge of processing the overwhelming amounts of sensory inputs from the external environment, the brain must allocate attentional resources to prioritize the processing of task-relevant information. Importantly, humans can proactively prepare for stimulus selection before the arrival of sensory inputs (*Summerfield and de Lange, 2014*). For example, when preparing to hail a taxi on the road, we tend to form a mental representation of the defining features of a taxi (e.g., yellow with a car-like shape). This ability relies on the formation of attentional templates – mental representations of the target – to accelerate stimulus selection and resolve perceptual competition by enhancing task-relevant information and suppressing irrelevant information (*Desimone and Duncan, 1995*; *Kastner et al., 1999*). While most attentional models posit that attentional templates during stimulus processing reflect veridical representations of the target (*Jigo et al., 2018*; *Malcolm and Henderson, 2009*), the nature of the template during preparation remains less understood.

A classical view suggests that attentional template during preparation may reflect veridical target features, analogous to the representational format during stimulus selection. However, evidence supporting this account has been mixed. For example, while some previous functional magnetic resonance imaging (fMRI) studies have demonstrated that preparatory activity contains target information similar to the sensory responses to the corresponding targets (*Kok et al., 2014*; *Lewis-Peacock et al., 2015*; *Stokes et al., 2009*), more recent electrophysiological studies suggest that, if anything, this template is engaged only shortly before the expected arrival of sensory input rather than being continuously active (*Grubert and Eimer, 2018*; *Myers et al., 2015*). Notably, in some cases, the template is even largely undetectable during preparation (*Wen et al., 2019*). Alternatively, an emerging view suggests a non-veridical template suffices for guiding attention during preparation, where precise processing of stimuli may be unnecessary at this stage. Support for this notion comes from the identification of attentional signals during preparation that differ from neural signals observed during perceptual target processing (*Gong et al., 2022*). Recent theories of visual search also propose a non-veridical, 'good-enough' template for early attentional guidance (*Wolfe, 2021*; *Yu et al., 2023*). However, it remains unclear whether a 'good-enough' template for search also applies to preparatory attention.

The notion that there may be a sensory and non-sensory attentional template might not be as far-fetched as it seems. Indeed, it is feasible that during preparation, following stimulus presentation, attentional signals undergo a transformation from a non-sensory to a sensory-like template. Previous behavioral (*Hamblin-Frohman and Becker, 2021*; *Yu et al., 2022*) and neural studies (*Gong et al., 2022*; *Jigo et al., 2018*; *Wen et al., 2019*) are generally consistent with this idea of coarse-to-fine transitions, suggesting that during preparation, a sensory-like template may not be initially necessary but only becomes relevant when the stimulus needs to be identified. However, if and in what way the brain coordinates these non-sensory and sensory-like templates remains unclear. Here, we propose that during preparation, a sensory-like template may be stored in a latent (e.g., activity–silent) state concurrently with a non-sensory template. This idea parallels recent findings from working memory studies, which suggest that information intended for proactive use is kept in activity–silent traces to support future behavior (*Stokes, 2015*; *Wolff et al., 2015*; *Wolff et al., 2017*). The present study seeks to determine the possibility of the latent, sensory-like template during the preparation for discriminating an upcoming stimulus.

To test these hypotheses, participants engaged in a cueing task in which they prepared during an extended period of time for the presentation of a compound stimulus grating containing the cued orientation and a distractor orientation. In addition, in order to be able to construct the sensory-format representations (leftward and rightward orientation), single orientations were presented during the perception task. Critically, we used a 'pinging' technique combined with multivariate decoding methods, which has been shown to be effective in retrieving information from latent brain states (*Duncan et al., 2023*; *Wolff et al., 2015*; *Wolff et al., 2017*; *Zhang and Luo, 2023*). In the standard condition (*No-Ping session*), the preparation period was devoid of visual impulses. During preparation, the neural activity patterns in visual and frontoparietal areas could discriminate between the orientations that participants were preparing for. Yet, neural activity patterns evoked by the preparation for upcoming orientations were distinct from those evoked by perception of orientations, suggesting a predominantly non-sensory template during preparation. By contrast, when we presented a high-contrast, task-irrelevant impulse stimulus during preparation (*Ping session*), neural activity patterns

activated by the preparation for orientation in the visual cortex were similar to those evoked by the perception of orientations, suggesting the existence of a latent, sensory-like format of representation during preparatory attention. Furthermore, the emergence of sensory-like templates coincided with enhanced information connectivity between V1 and frontoparietal areas and was associated with improved behavioral performance. Our findings provide evidence for the co-existence of two formats of attentional templates (non-sensory vs. sensory-like) during preparation, as well as a novel neural mechanism for their maintenance in different functional states (active vs. latent). We propose that this dual-format representation may serve to increase flexibility of attentional control.

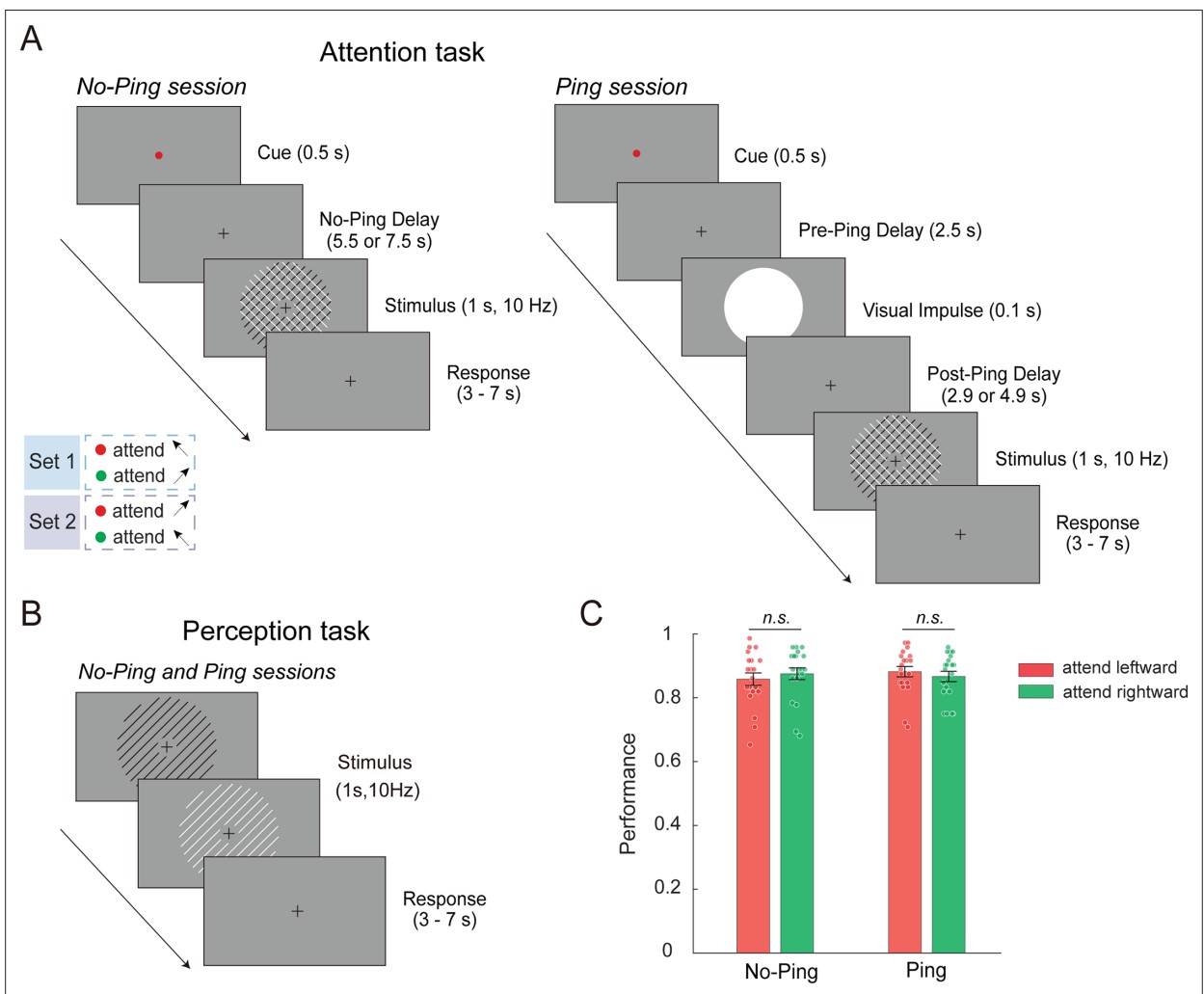

**Figure 1.** Experiment procedure and behavioral performance. (**A**) Attention tasks in the No-Ping and Ping sessions. Note that only long-delay trials are shown. A small proportion of short-delay trials (20%, with a delay of 1.5 or 3.5 s) were included to create temporal uncertainty and encourage consistent active preparation during the delay. Both component gratings were flickering at 10 Hz between white and black, so that luminance could not confound either the task strategy (e.g., attending to luminance) or neural measures. The inset shows two sets of color-orientation mapping, which were reversed halfway through the experiment to minimize the impact of cue-induced sensory difference on neural activity. A high-contrast impulse was presented during the preparation period in the Ping session. (**B**) Perception task. Similar to the attention task, the single-orientation grating also flickered at 10 Hz between white and black. (**C**) Behavioral accuracy in the attention tasks in the No-Ping and Ping sessions. Each dot represents one subject's data. Error bars denote standard error of the means (SEM).

The online version of this article includes the following source data for figure 1:

**Source data 1.** Behavioral accuracy in the attention tasks.

## Results

### Behavioral performance during the attention tasks

In an orientation cueing task, participants were shown a color cue indicating the reference orientation (45° or 135°) to attend to during preparation period (a delay of 5.5 or 7.5 s) with or without the impulse perturbation (*Figure 1A*). This was followed by the presentation of a compound stimulus consisting of two oriented gratings. During the stimulus selection period (after the gratings appeared), participants were tasked with discriminating a small angular offset of the cued grating from the cued reference orientation. The angular offset was individually thresholded before the scanning sessions (mean offset = 2.50° in the No-Ping session and 2.52° in the Ping session) without significant difference between the two sessions (independent *t*-test: $t(38) = 0.085$, p = 0.932, Cohen's $d = –0.027$). Participants' discrimination performance showed no significant difference between two attended orientations in either the No-Ping (paired *t*-test: $t(19) = 1.439$, p = 0.166, Cohen's $d = –0.321$) or the Ping session (paired *t*-test: $t(19) = 0.494$, p = 0.627, Cohen's $d = 0.122$; *Figure 1C*). A two-way mixed ANOVA (attended orientation × session) revealed neither significant main effects (attended orientation: $F(1,38) = 0.392$, p = 0.535, $\eta_p^2 = 0.01$; session: $F(1,38) = 0.001$, p = 0.970, $\eta_p^2 < 0.001$) nor interaction effect ($F(1,38) = 1.811$, p = 0.186, $\eta_p^2 = 0.045$). Bayesian analyses provided moderate evidence to support the null hypothesis ($BF_{excl} > 3.633$), suggesting comparable performance levels between two sessions and two attended orientations.

### A default, non-sensory representation of attentional template during preparation

The first aim of this study was to determine whether attentional signals during preparation are encoded in a sensory-like or non-sensory format. To address this, we first examined whether, in the attention task during the No-Ping condition, the distributed neural pattern contained feature-specific information. We trained and tested separate classifiers to predict the attended orientation during the preparation and stimulus selection periods (*Figure 2A*, 'Attention decoding'; see Materials and methods for details). This analysis was performed for each of the four regions along the visual hierarchy, including primary visual cortex (V1), extrastriate visual cortex (EVC), intraparietal sulcus (IPS), and prefrontal cortex (PFC). The average decoding accuracies for both preparation and stimulus selection periods were significantly above-chance level in each region (permutation analyses: ps < 0.004 across regions, *Figure 2B*), indicating that the brain maintained reliable information about the attended feature both before and after the onset of the compound grating. Next, we examined whether the preparatory activity reflected a sensory-like format of attentional template (*Figure 2A*, 'Cross-task generalization', see Materials and methods). We trained a classifier using data from the perception task (leftward vs. rightward orientation; *Figure 1B*) and tested its performance on data from the preparation period in the attention task (attend leftward vs. attend rightward). However, this cross-task generalization analysis yielded no significant effects (ps > 0.132 across the regions). In contrast, we observed above-chance generalization from the perception task to the stimulus selection period (ps < 0.001 across regions, *Figure 2C*), confirming previous findings of the sensory-like attentional template following stimulus presentation (*Gong et al., 2022*; *Jigo et al., 2018*; *Wen et al., 2019*).

Before drawing conclusions based on the lack of generalization from the perception task to preparatory attention, we considered two alternative explanations to rule out potential confounds. First, the robust attention decoding during preparation ruled out the possibility that participants were not actively engaged in the task during preparation (*Figure 2B*, unfilled bars). Second, the generalizable effect from the perception task to the stimulus selection period across regions (*Figure 2C*, filled bars) argues against the possibility of low statistical power. Overall, these findings suggest that the preparatory attention and sensory processing of features have distinct formats, presumably reflecting a non-sensory format of representation during the preparation. These results replicate those of a previous fMRI study using motion stimuli with a similar design (*Gong et al., 2022*). Furthermore, consistent with previous studies (*Gong et al., 2022*; *Jigo et al., 2018*), univariate analysis did not reveal any reliable difference in overall BOLD responses between attention orientations (*Appendix 1—figure 1*).

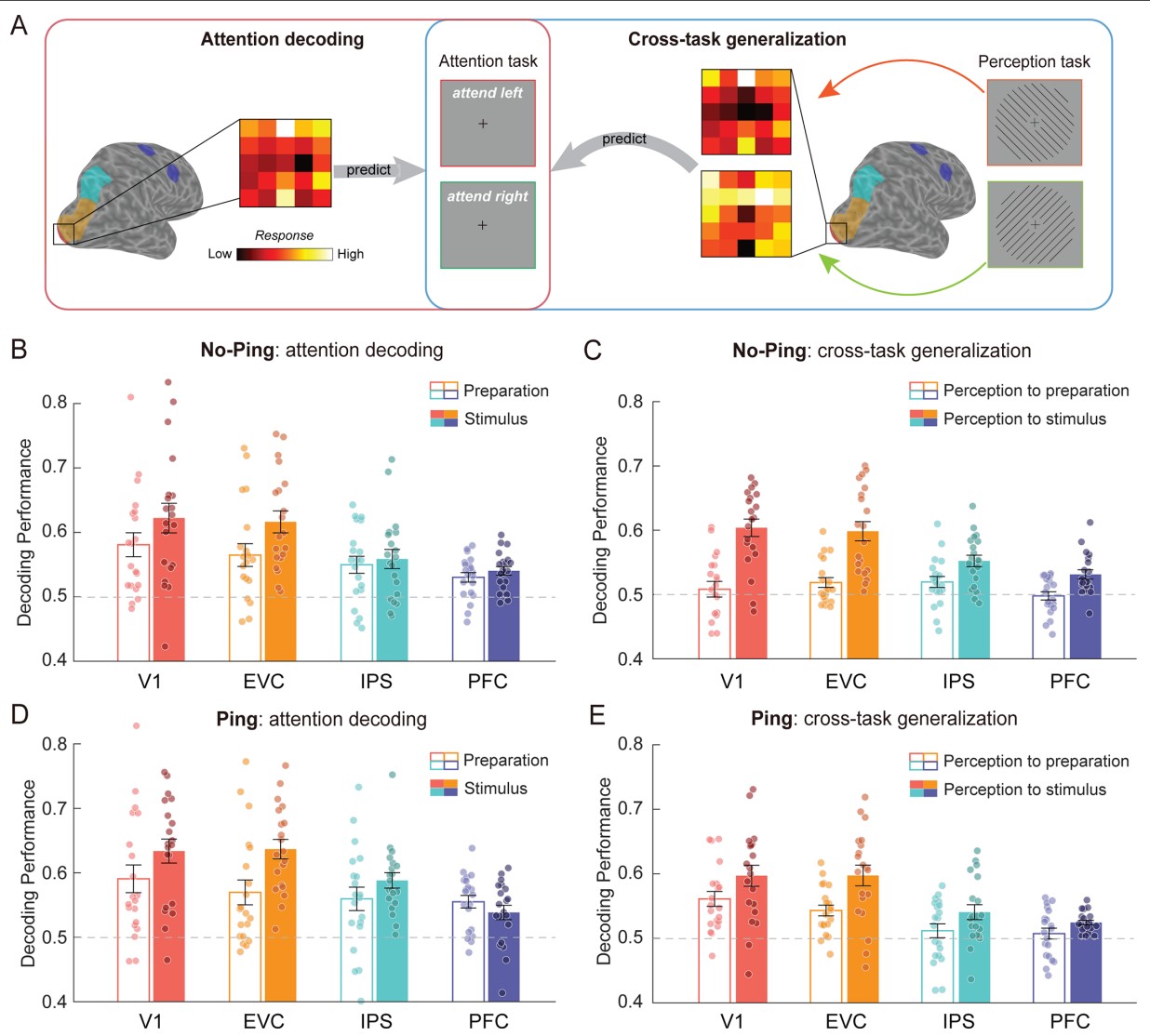

**Figure 2.** Multivoxel pattern analysis (MVPA) for the No-Ping session and Ping session. (**A**) Schematic illustration of the decoding of attended orientation (attend leftward vs. attend rightward) in the attention task (left panel) and the cross-task generalization analysis from perception task to the attention task (right panel). The four regions are shown on a representative right hemisphere as colored areas: V1 is marked in red, extrastriate visual cortex (EVC) in yellow, intraparietal sulcus (IPS) in cyan, and prefrontal cortex (PFC) in purple. (**B**) Decoding accuracy during preparation and stimulus selection periods across regions in the No-Ping and (**D**) Ping session. (**C**) Cross-task generalization performance from the perception task to the preparatory periods and the stimulus selection periods across regions in the No-Ping and (**E**) Ping session. The dashed lines represent the theoretical chance level (0.5). Each dot represents one subject's data. Error bars denote SEM.

The online version of this article includes the following source data and figure supplement(s) for figure 2:

**Source data 1.** Decoding accuracy across brain regions in the attention tasks.

**Source data 2.** Cross-task decoding generalization across brain regions.

**Figure supplement 1.** Sensory decoding in the perception task.

**Figure supplement 1—source data 1.** Decoding accuracy across brain regions in the perception task.

## A latent, sensory-like attentional template during preparation revealed by visual impulse

The second aim of our study was to examine whether a latent, sensory-like template exists during preparation. While this precise template may not be necessary for preparation, it is relevant for subsequent target selection and discrimination (i.e., select the cued grating from the compound stimulus

and discriminate a small angular offset between the cued grating and the reference orientation). To test this hypothesis, we perturbed the neural activity by means of a visual impulse during the preparation period in the Ping session (*Figure 1A*, right panel). Using the same analyses as those performed in the No-Ping session, robust attentional signals were observed during both preparation and stimulus selection periods (permutation analyses: ps < 0.001 across regions; *Figure 2D*). Importantly, the cross-task generalization analyses indicated that the visual impulse led to above-chance generalization from the perception task to preparation period (*Figure 2E*, unfilled bars) in V1 and EVC (ps < 0.001), but not in IPS and PFC (ps > 0.584), along with generalizable effects from the perception task to the stimulus selection periods (ps < 0.036 across regions; *Figure 2E*; filled bars). These results suggest that different brain areas are involved in coding for sensory-like templates. To further evaluate whether the cross-task generalization from the perception task to the preparation period was statistically different with and without visual impulse, we conducted a two-way mixed ANOVA (session × region) on the generalization performance. The analysis revealed main effects of region ($F(3,114) = 5.220$, p = 0.002, $\eta_p^2 = 0.121$), session ($F(1,38) = 7.321$, p = 0.010, $\eta_p^2 = 0.162$), and importantly, a significant interaction effect ($F(3,114) = 3.964$, p = 0.010, $\eta_p^2 = 0.094$) that the visual impulse led to significantly increased decoding accuracy in V1 (independent $t$-test: $t(38) = 3.145$, p = 0.003, Cohen's $d = 0.995$) and EVC (independent $t$-test: $t(38) = 2.153$, p = 0.038, Cohen's $d = 0.681$), but not in the frontoparietal regions (ps > 0.374). This dissociable result between the two sessions further supports the activation of a latent, sensory-like template by the visual impulse during preparatory attention.

To further solidify this conclusion, the following analyses were used to examine several alternative possibilities. First, we examined whether the impulse-driven generalization resulted from stronger feature information in the Ping compared to No-Ping session during the perception task. This was not the case, as evidenced by comparable levels of decodable orientation information between the Ping and No-Ping sessions (see *Figure 2—figure supplement 1*). Next, we asked whether the increased generalization was due to generally stronger attentional signals in the Ping session during the attention tasks – for example, if visual impulses simply refocused attention during long delays. This was not the case, as the two-way mixed ANOVAs (session × region) on attention decoding accuracy revealed neither a significant main effect of session nor an interaction effect during both the preparation (ps > 0.519; $BF_{excl} > 3.247$) and stimulus selection periods (ps > 0.336; $BF_{excl} > 3.297$), suggesting comparable amount of attentional information between the two sessions. Therefore, the findings of impulse-driven sensory-like template in the visual cortex during preparation cannot be explained by general differences between two sessions.

## Matching preparatory attention to sensory template: impact on neural representation and behavior

The reported decoding accuracy from the cross-task generalization analysis quantifies the degree to which differences in neural activity pattern between two conditions are shared across attention and perception tasks. However, it does not directly measure how similar the neural patterns are when attending to an orientation compared to perceiving that orientation. Unlike decoding accuracies, Mahalanobis distance provides a continuous measure for characterizing representational geometries between different conditions (*Mahalanobis, 1936*). To further corroborate our findings of the impulse-driven sensory-like template, we calculated the Mahalanobis distance between each attention condition during preparation and each perception condition (see Materials and methods). If the patterns of activity reflect a sensory-like template, we would expect greater pattern similarity (smaller distance) between 'attend leftward' and 'perceive leftward' than between 'attend leftward' and 'perceive rightward', and vice versa for the 'attend rightward' conditions (see *Figure 3A* for a schematic of the four pair-wise distance measures), leading to an interaction between attended and perceived orientation conditions.

We used a two-way repeated-measures ANOVA (attended orientation × perceived orientation) on the Mahalanobis distance, separately for each session and each region. During the preparatory period in the No-Ping session, no significant interaction effects were observed across regions (ps > 0.443; *Figure 3B*). In contrast, the same analyses applied to the Ping session revealed significant interaction effects in visual areas (V1: $F(1,19) = 9.335$, p = 0.007, $\eta_p^2 = 0.329$; EVC: $F(1,19) = 8.563$, p = 0.009, $\eta_p^2 = 0.311$; *Figure 3C*), but not for frontoparietal regions (ps > 0.213). This cross-region difference is consistent with the function of sensory areas in encoding precise neural representations for basic

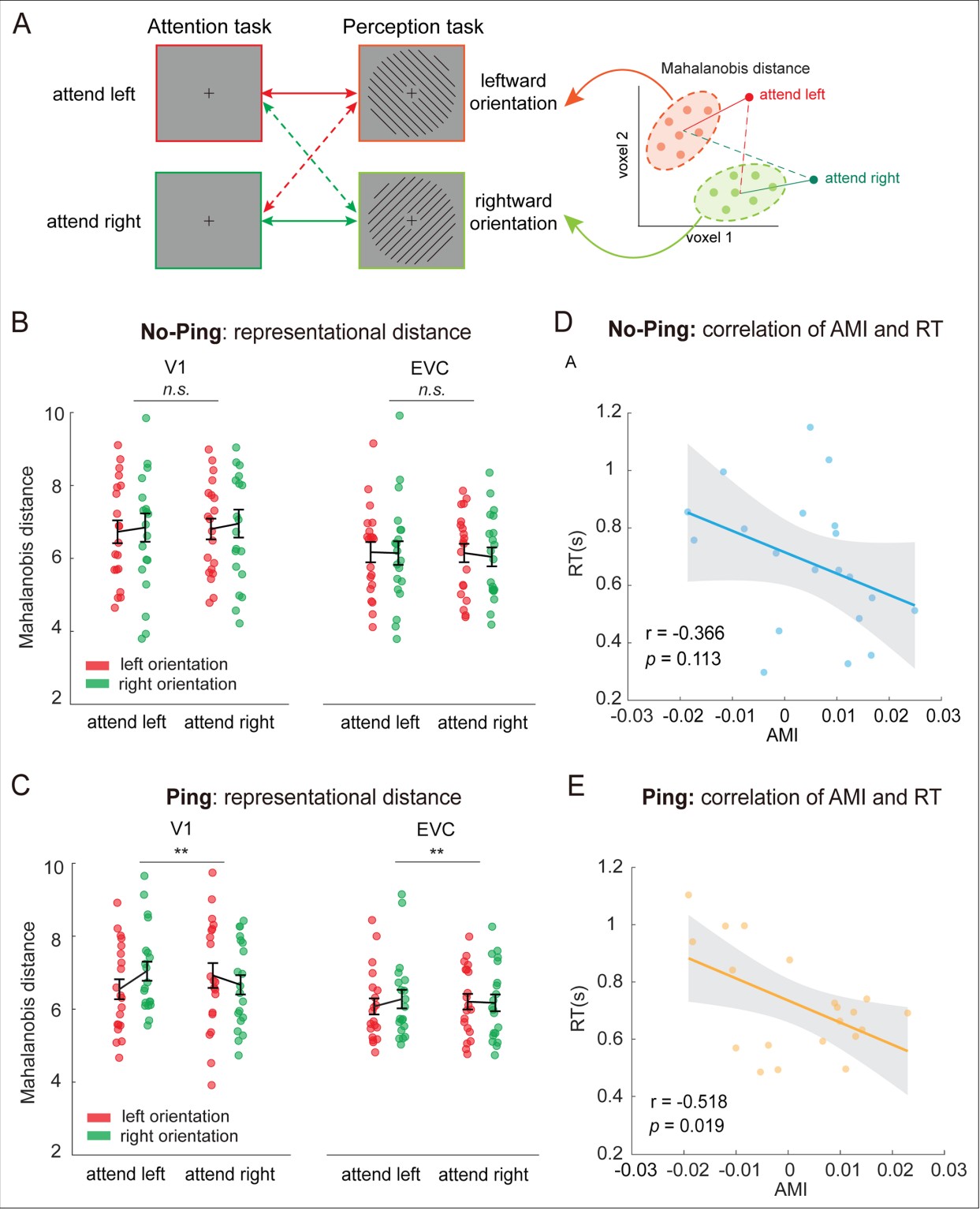

**Figure 3.** Orientation-selective attentional modulations on neural pattern distances during preparation. (**A**) Schematic illustration of the representational distance (mean Mahalanobis distances) between each of the attention conditions and each of the perception conditions. Colored arrows indicate measures of the pair-wise Mahalanobis distance. The right panel shows two attention trials (red indicates attend-to-leftward and green indicates attend-to-rightward) to the distribution of each perception condition (shown in a cloud of light-colored dots). (**B**) Mahalanobis distance between preparatory attention condition and perceived orientation condition in the No-Ping and (**C**) Ping sessions. Error bars denote SEM. (**D**) Correlations between

*Figure 3 continued on next page*

*Figure 3 continued*

attentional modulation index (AMI) and reaction time (RT) in the No-Ping and (**E**) Ping sessions. Each dot represents one subject's data. The shaded area represents the confidence intervals of the regressed lines. **p < 0.01.

The online version of this article includes the following source data and figure supplement(s) for figure 3:

**Source data 1.** Mahalanobis distances between preparatory attention conditions and the perception conditions.

**Source data 2.** Attentional modulation index and reaction time.

**Figure supplement 1.** Mahalanobis distance during stimulus selection period.

**Figure supplement 1—source data 1.** Mahalanobis distances between stimulus-based attention conditions and the perception conditions.

**Figure supplement 2.** Relationship between attentional modulation on Mahalanobis distance and RT.

**Figure supplement 2—source data 1.** Reaction time in strong and weak attentional modulation trials.

visual features. Next, we directly compared whether attentional modulation of Mahalanobis distance was statistically different with and without the visual impulse. We defined a new condition label based on orientation consistency between attended and perceived orientations: (1) same orientation: averaging 'attend leftward/perceive leftward' and 'attend rightward/perceive rightward'; and (2) different orientation: averaging 'attend leftward/perceive rightward' and 'attend rightward/perceive leftward'. A two-way mixed ANOVA (session × orientation consistency) on Mahalanobis distance revealed a main effect of orientation consistency in V1 ($F(1,38) = 4.21$, $p = 0.047$, $\eta_p^2 = 0.100$), indicating that activity patterns were more similar when attended and perceived orientations matched. No significant main effect of session was found ($p = 0.923$). Importantly, a significant interaction was found in V1 ($F(1,38) = 5.00$, $p = 0.031$, $\eta_p^2 = 0.116$), suggesting that visual impulse enhanced the similarity between preparatory attentional template and the perception of corresponding orientation. In EVC, the same analysis revealed only a main effect of orientation consistency ($F(1,38) = 5.87$, $p = 0.020$, $\eta_p^2 = 0.134$), with no other significant effects (ps >0.240). We also calculated the Mahalanobis distance between neural patterns evoked by superimposed gratings during the stimulus selection period and each condition in the perception task, finding similar results (*Figure 3—figure supplement 1*). This result was expected, as feature-based attention is known to selectively enhance task-relevant features while filtering out task-irrelevant ones.

The continuous nature of the Mahalanobis distance also made it possible to further investigate potential neural–behavioral correlations. We examined whether activating a sensory-like template during preparation would benefit subsequent orientation processing. In particular, we calculated attentional modulation indices (AMIs) based on trial-wise Mahalanobis distance in V1. The index was calculated as follows: AMI = $(D_{different} - D_{same})/(D_{different} + D_{same})$, where $D_{same}$ and $D_{different}$ are the measured distance ($D$) in the Same (e.g., attend and perceive the same orientation) and Different (e.g., attend and perceive different orientations) orientation condition, respectively (see Methods and materials). Then, we calculated the correlation between AMI and both reaction time (RT) and accuracy across participants, separately for each session. In the No-Ping session, we observed no significant correlation between AMI in V1 and RT ($r = -0.366$, $p = 0.113$; *Figure 3D*). By contrast, the same analysis in the Ping condition revealed a significantly negative correlation ($r = -0.518$, $p = 0.019$; *Figure 3E*). These results indicate that the attentional modulations evoked by visual impulse were associated with faster RTs. These effects were not observed for accuracy (ps > 0.550). Furthermore, we also performed within-subject analysis by sorting trials as 'strong modulation' and 'weak modulation' trials based on each individual's AMI values, facilitated RTs were observed in 'strong modulation' trials during the Ping session (*Figure 3—figure supplement 2*). These results suggest that the impulse-driven sensory-like template in primary visual cortex is functionally relevant to subsequent attentional selection, providing evidence for the prospective use of sensory-like template in this task. In addition, we did not observe such behavioral differences in analogous analyses using data from the stimulus selection period in either session (ps > 0.230), which might be due to the potential dilution by strong stimulus-evoked responses during the stimulus selection period.

## Activating sensory-like template strengthens the informational connectivity between sensory and frontoparietal areas

Selective attention is generally believed to rely on coordinated network activity (*Corbetta and Shulman, 2002*). In particular, studies have shown that functional connectivity between sensory and frontoparietal areas was modulated by attentional control (*Bressler et al., 2008*; *Rosenberg et al., 2020*). Given that the impulse-driven sensory-like template facilitated behavior, we reasoned that it may also enhance network communication. Thus, we examined informational connectivity (IC) measures to explore how the impulse altered network function during the attention task.

We used a method that allows inference based on multivoxel pattern information rather than univariate BOLD response (*Jia et al., 2020*; *Ng et al., 2021*). For each region of interest (ROI), we calculated the cross-validated Mahalanobis distance from each attention trial (from one left-out run) to the distribution of each attended orientation (all trials from remaining runs) during preparation (see Methods and materials). To quantify the degree of attentional modulation during preparation, we calculated the AMI based on trial-wise Mahalanobis distance and generated a time course of AMI values across trials (see *Figure 4A* for the schematic). Pearson correlation was used to estimate the covariation between each pair of ROIs, and the resulting correlation coefficients were transformed using Fisher's *z*-transform for statistical inference (*Figure 4B*). The analysis revealed numerically higher levels of connectivity in Ping than in No-Ping session. This impulse-driven increase in connectivity reached statistical significance in two pairs (*Figure 4C*): V1–IPS (independent *t*-test: $t(38) = 2.566$, $p = 0.014$; Cohen's $d = 0.812$) and V1–PFC (independent *t*-tests: $t(38) = 3.158$, $p = 0.003$; Cohen's $d = 0.999$). The enhanced functional connectivity between V1 and frontoparietal areas driven by the impulse may potentially facilitate information flow among areas to improve attentional control, as implicated by a trend of ping-enhanced correlations between V1-PFC and RTs (*Figure 4—figure supplement 1*). Additionally, the same analysis of AMI based on cross-validated Mahalanobis distance during the stimulus selection period showed no significant differences in information connectivity between No-Ping and Ping sessions (ps >0.224; *Figure 4—figure supplement 2*). The lack of changes in long-range connectivity during the stimulus selection period may be attributed to a general rise in connectivity caused by strong sensory inputs in this period, which could have attenuated any potential impacts of visual impulses. Furthermore, connectivity analysis based on mean BOLD response over time did not reveal any significant changes in inter-cortical connections between the two sessions (ps > 0.136; *Figure 4—figure supplement 3*), suggesting that the impulse-driven increased information connectivity between V1 and higher-order areas was unlikely contributed by the overall changes of BOLD response.

## Discussion

While there is ample evidence that the brain can maintain an attentional template of an upcoming target before sensory information is presented (*Desimone and Duncan, 1995*; *Kastner et al., 1999*; *Summerfield and de Lange, 2014*), its representational format remains unclear. To address this, we used an orientation cueing paradigm with separated preparation and stimulus selection periods and applied multivoxel pattern analysis (MVPA) to decode neural activity patterns associated with feature-specific attentional information of the upcoming target. The analyses showed robust attentional information both before and after the presentation of the compound grating, indicating a sustained maintenance of attentional templates throughout a trial. Importantly, while the decoders trained on the perception of single orientations could not generalize to preparation until the stimulus selection period (*No-Ping session*), perturbing the brain with a visual impulse resulted in generalizable activity patterns during preparation in V1 and EVC (*Ping session*). These results suggest a predominantly non-sensory format of representation, with a sensory-like template in a latent state during feature-based preparation in the visual cortices. Furthermore, impulse-driven sensory-like template was accompanied by enhanced information connectivity between V1 and frontoparietal areas, as well as enhanced orientation-specific neural modulations of neural distances in the visual areas that predicted levels of behavioral performance. We observed similar response profiles in V1 and EVC, with V1 exhibiting more robust ping-evoked changes compared to EVC, consistent with its primary role in orientation processing (*Priebe, 2016*). The differences between the Ping and No-Ping sessions could not be attributed to differences in sensory information from the perception task, overall strength of

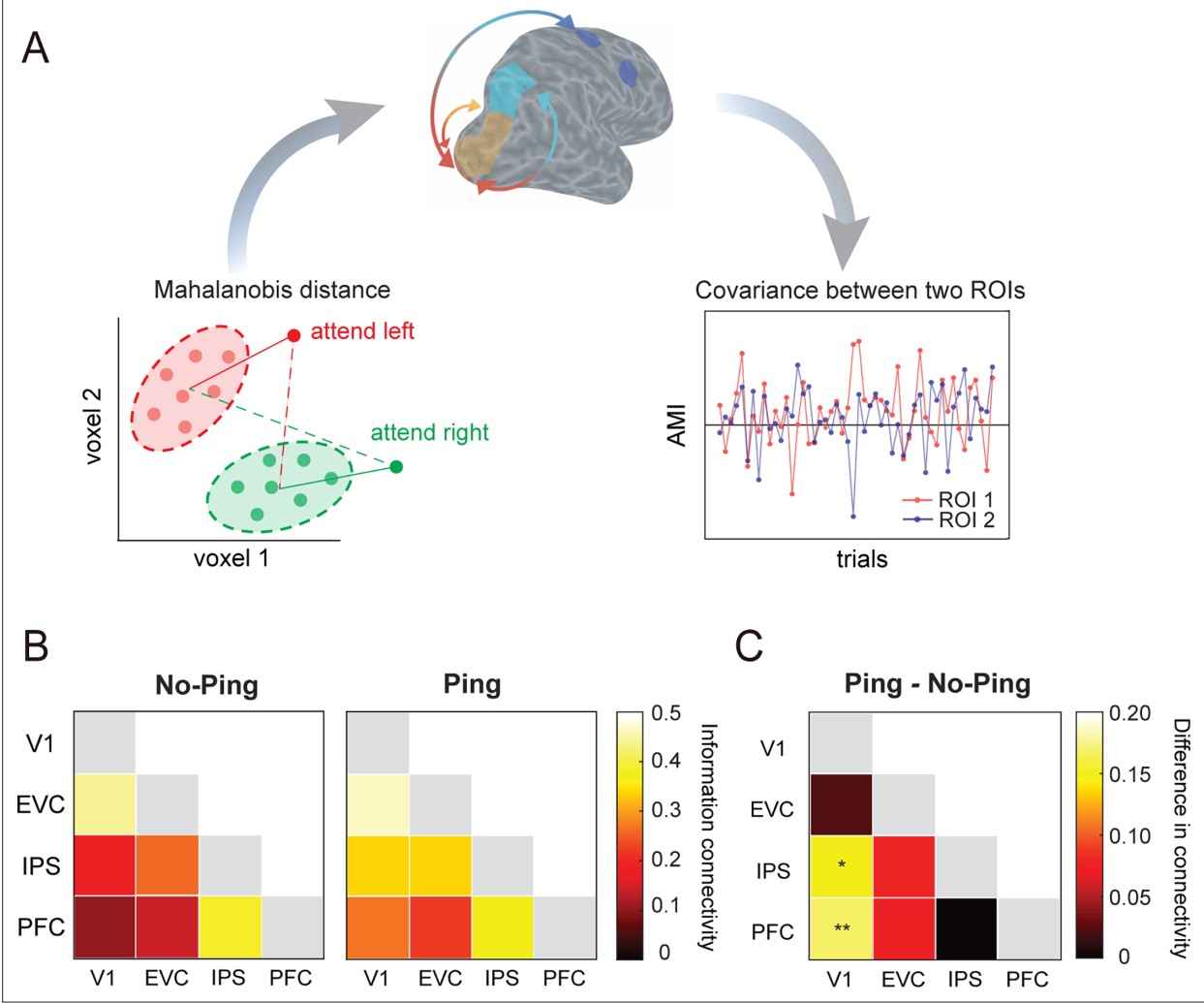

**Figure 4.** Information connectivity analysis. (**A**) Schematic illustration of the procedure for the information connectivity analysis in the space of two hypothetical voxels. For each region, we calculated the Mahalanobis distance of the attention trial (from one left-out run) from two attention distributions (all trials from remaining runs). Red and green dots indicate activity patterns from two trials (right panel). The brain image shows an example pair of intercortical information connectivity between V1 and prefrontal cortex (PFC). The time series (lower-left panel) consisted of attentional modulation index (AMI) based on the Mahalanobis distance. (**B**) Between-region information connectivity in the No-Ping and Ping sessions. (**C**) The differences in connectivity between the Ping and No-Ping sessions. *p < 0.05, **p < 0.01.

The online version of this article includes the following source data and figure supplement(s) for figure 4:

**Source data 1.** Information connectivity between brain regions during the preparation period.

**Figure supplement 1.** Relationships between information connectivity and reaction time.

**Figure supplement 1—source data 1.** Information connectivity in prefrontal cortex (V1–PFC) during preparation and reaction time.

**Figure supplement 2.** Functional connectivity analysis based on multivoxel activity patterns during the stimulus selection period.

**Figure supplement 2—source data 1.** Information connectivity between brain regions during the stimulus selection period.

**Figure supplement 3.** Functional connectivity analysis based on mean BOLD activity during the preparation period.

**Figure supplement 3—source data 1.** Information connectivity between brain regions during the preparatory period based on BOLD activity.

preparatory attention, or differences in eye position (*Appendix 1—figure 2*). Therefore, our findings suggest a dual-format neural representation scheme (non-sensory vs. sensory-like) operating in different functional states (active vs. latent). This mechanism may give rise to flexible attentional control, allowing effective transition from coarse to fine attentional templates at various processing stages (initial guidance vs. precise stimulus discrimination).

Recent advances in theories of visual search differentiated between the 'guiding template' and 'target template' based on measures of behavioral performance and eye movements (*Wolfe, 2021*; *Yu et al., 2023*). According to these theories, early attentional guidance typically depends on non-veridical codes that represent only the most diagnostic information, whereas later, target-match processes utilize more precise codes to optimize decision accuracy (*Kerzel, 2019*; *Scolari et al., 2012*; *Yu et al., 2022*). Our study reveals a parallel coding mechanism in the context of feature-based attention, expanding upon these theoretical notions in three key aspects.

First, we provide neural evidence for a default, predominantly non-sensory template during preparation, indicating that the concept of a 'guiding template', as proposed by visual search theories (*Wolfe, 2021*; *Yu et al., 2023*), also applies to preparatory attention in a non-search context. This highlights a shared functional role of a non-veridical attentional template in early guidance across different scenarios. Second, despite the theoretical notion that the brain maintains a more veridical template with detailed target information than is typically utilized to form the guiding template (*Wolfe, 2021*; *Yu et al., 2023*), neural evidence supporting this hypothesis is currently lacking. We provide evidence for this notion and propose a plausible neural implementation for preserving a more veridical, sensory-like template in the latent state. A natural question is why the sensory-like template remains latent during preparation. We note that our task requires both coarse and fine featural information. During preparation, a coarse, non-veridical guiding template suffices for target-distractor discrimination, while during stimulus selection, a precise template is needed for the fine discrimination task (reporting the tilted direction of a small angular offset). Maintaining a latent sensory-like template during preparation is thus efficient, as it facilitates future sensory processing while conserving resources. Finally, information connectivity between visual and higher-order frontoparietal regions was enhanced by visual impulse during preparation, which correlated with improved behavioral performance in feature selection. This result suggests that improved information flow across the relevant areas leads to enhanced attentional control, which in turn contributes to refined sensory representations of target in early visual cortex, facilitating transitions from a non-sensory to sensory-like template. Future studies may adopt layer-specific fMRI to infer the direction of this improved information flow (*Jia et al., 2023*; *Jia et al., 2024*) and explore the relationship between long-range connections and the utilization of different formats of target templates.

It could be argued that preparatory attention relies on the same mechanisms as working memory maintenance (*Bettencourt and Xu, 2016*; *Sheremata et al., 2018*). While these functions are intuitively similar and likely overlap, there is also evidence indicating that they can be dissociated (*Battistoni et al., 2017*). In particular, we note that in our task, attention is guided by symbolic cues (color-orientation associations), while working memory tasks typically present the actual visual stimulus as the memorandum. A central finding in working memory studies is that neural signals during WM maintenance are sensory in nature, as demonstrated by generalizable neural activity patterns from stimulus encoding to maintenance in visual cortex (*Harrison and Tong, 2009*; *Serences et al., 2009*; *Rademaker et al., 2019*). However, in our task, neural signals during preparation were non-sensory, as demonstrated by a lack of such generalization in the No-Ping condition (see also *Gong et al., 2022*). We believe that the differences in cue format and task demand in these studies may account for such differences. In addition to the difference in the sensory nature of the preparatory versus delay-period activity, our ping-related results also exhibited divergence from working memory studies (*Wolff et al., 2015*; *Wolff et al., 2017*). While these studies used the visual impulse to differentiate active and latent representations of *different items* (e.g., attended vs. unattended memory item), our study demonstrated the active and latent representations of *a single item in different formats* (i.e., non-sensory vs. sensory-like). Moreover, unlike our study, the impulse did not evoke sensory-like neural patterns during memory retention (*Wolff et al., 2017*). These observations suggest that the cognitive and neural processes underlying preparatory attention and working memory maintenance could very well diverge. Future studies are necessary to delineate the relationship between these functions both at the behavioral and neural level.

While we found that the ping allowed us to detect a sensory-like template during preparation, the underlying neural mechanism of such effects remains unclear. One possibility, as informed by theoretical studies of working memory, is that the sensory-like template could be maintained via an 'activity-silent' mechanism through short-term changes in synaptic weights (*Mongillo et al., 2008*). In this framework, a visual impulse may function as nonspecific inputs that momentarily convert latent

traces into detectable activity patterns (*Rademaker and Serences, 2017*). Related to our findings, it is unlikely that the orientation-specific templates observed during the Ping session emerged de novo from purely non-sensory representations and were entirely induced by an exogenous ping, which was devoid of any orientation signal. Instead, the more parsimonious explanation is that visual impulse reactivated pre-existing latent sensory signals, consistent with the models of 'activity-silent' working memory. However, the detailed circuit-level mechanism of such reactivation is still unclear, as well as whether this effect is modality specific. Prior work shows that only visual, but not auditory, impulses reactivate latent visual working memory (*Wolff et al., 2020*), suggesting some degree of modality specificity. However, this finding warrants direct investigation in future studies. Furthermore, we acknowledge that whether pinging identifies an activity-silent mechanism is currently debated (*Barbosa et al., 2021*; *Schneegans and Bays, 2017*). An alternative possibility is that the visual impulse amplified a subtle but active representation of the sensory template during preparation. Distinguishing between these alternatives likely requires future studies with more detailed neurophysiological measurements. Regardless of the precise neural mechanism for the observed latent, sensory representation, our results suggest that both sensory and non-sensory templates likely co-exist.

The non-generalizable activity patterns from perception to preparatory attention, in the absence of visual impulse, suggest a default, predominantly non-sensory template during preparation. This finding is largely consistent with electrophysiological studies (*Myers et al., 2015*; *Wen et al., 2019*) and our prior fMRI work on preparatory attention to motion directions (*Gong et al., 2022*), but differs from some previous neuroimaging studies that demonstrated sensory-like templates during preparation (*Kok et al., 2014*; *Peelen and Kastner, 2011*; *Stokes et al., 2009*). One potential account for these discrepancies is that those studies used cue-only trials where the target was expected but not actually presented, in contrast to our task where the target was shown on every trial with temporally separated preparation and stimulus selection periods. This seemingly subtle difference may significantly impact the formats of the neural representations. Because cue-only trials increased the likelihood of target appearance at the subsequent time point, sensory template may be activated due to modulations of temporal expectations (*Grubert and Eimer, 2018*). This explanation is consistent with theories suggesting differential influences of expectation and attention on neural activity: expectation reflects visual interpretations of stimuli due to sensory uncertainty, whereas attention is guided based on the task relevance of sensory information (*Rungratsameetaweemana and Serences, 2019*; *Summerfield and Egner, 2009*; *Summerfield and Egner, 2016*). Our finding of a predominantly non-sensory format may indicate an optimized coding strategy employed by the brain to effectively and robustly represent information for future use. This aligns with the proposed role of attention in modulating sensory representations to encode only currently relevant information at a minimal cost (*Yu et al., 2023*).

While our findings cannot pinpoint the exact format of this non-sensory template, we consider categorical coding a plausible candidate based on previous findings. For instance, visual search studies demonstrate that categorical attributes (e.g., steep vs. shallow; left-tilted vs. right-tilted) efficiently guide attention for simple features, such as an orientation or a color (*Kong et al., 2017*; *Wolfe et al., 1992*), particularly when features are consistent and predictable (*Hout et al., 2017*). In our task, the angular relations between the target and distractor orientation were defined by categorical attributes (e.g., left-tilted vs. right-tilted) and remained consistent across trials, making a categorical template feasible during preparatory attention. Furthermore, the categorical template allows for greater tolerance of stimulus variability, which is also useful given the trial-by-trial variations in target orientation around the reference orientation in our task. Future studies are needed to address the nature of the non-sensory template during preparation as well as task parameters that might modulate them.

In summary, the current study suggests that there are two formats of attentional templates, each having a distinct functional state: a default, non-sensory format and a latent, sensory-like format. This dual-format representation aligns with theories on the dual-function of attentional template for different task goals (*Hout and Goldinger, 2015*; *Yu et al., 2023*). The current findings provide a plausible neural implementation for these theories by demonstrating different formats in different functional states. This mechanism likely reflects an optimized coding scheme that effectively balances processing efforts and demands, particularly well suited for flexible control and transitions from coarse to fine task demands in visually guided behavior.

# Materials and methods

## Participants

Twenty individuals participated in the No-Ping session (11 females, mean age = 22.9) and twenty individuals participated in the Ping session (14 females, mean age = 23.7). Among them, 14 participants took part in both sessions, while 12 of them took part in only one session. The sample size was comparable to previous studies using similar attention tasks (*Baldauf and Desimone, 2014*; *Gong and Liu, 2020a*; *Gong and Liu, 2020b*; *Guo et al., 2012*; *Jigo et al., 2018*; *Liu and Hou, 2013*). Because our primary interest is the generalization from the perception task to the attention task, we used the minimal effect size of decoding accuracy across regions (one-sample *t*-tests: $d$=0.868) from our previous study with a similar design (*Gong et al., 2022*), and used G*Power (Version 3.1) (*Faul et al., 2007*) to confirm that this sample size is sufficient to detect a cross-task generalization effect with a power greater than 95% ($a$ = 0.05). All participants were right-handed and had a normal or corrected-to-normal vision. Participants provided written informed consent according to the study protocol approved by the Institutional Review Board at Zhejiang University (2020-06-001). They were paid ¥200 (~$27.4) for their participation in each session.

## Stimuli and apparatus

Stimuli were generated using Psychtoolbox (*Brainard, 1997*; *Kleiner et al., 2007*) implemented in MATLAB. The stimuli were presented on an LCD monitor (resolution: 1920 × 1080,, refresh rate: 60 Hz) during behavioral training, at a viewing distance of 90 cm in a dark room. During the fMRI scans, stimuli were projected to a screen via a MR-compatible LCD projector (PT-011, Jiexin Technology Co, Ltd, Shenzhen, China) with the same resolution and refresh rate as the LCD monitor during behavioral training. Participants viewed the screen via an angled mirror attached to the head coil at a viewing distance of 115 cm. Angular stimulus size was the same across behavioral and fMRI sessions.

The orientation stimuli were square-wave gratings (1.3 cycles per deg, duty cycle: 10%) in a circular aperture (inner radius: 1.5°; outer radius: 6°). The gratings flashed on a gray background at 10 Hz, alternating between black and white. There were two types of stimuli: two overlapping gratings oriented leftward (~135°) and rightward (~45°), or a single grating with one of the two orientations (~135° or ~45°). Here, we refer to the 45° and 135° orientations as the reference orientations. The impulse stimulus was a high-contrast, white (at the maximum projector output level) circular disk that covered the same area as the orientation stimulus (radius: 6°).

## Experimental procedures and tasks

Each participant completed at least two fMRI sessions on different days. One session was used for defining ROIs (see Definition of ROIs), while the remaining sessions were used for the main experiment (see Attention task and Perception task). Before the scanning sessions, participants were trained to familiarize themselves with the tasks in a separate behavioral session. The procedures and tasks were similar to our previous work (*Jigo et al., 2018*; *Gong et al., 2022*).

### Attention task

We used a cueing paradigm (*Figure 1A*). Each trial began with a color cue (red or green) for 0.5 s to indicate the reference orientation of the upcoming target (leftward vs. rightward orientation). In the No-Ping session, the cue was followed by a blank display during the preparation period; in the Ping session, a task-irrelevant, high-luminance visual impulse ('ping', 0.1 s) occurred at either 0.5 s (for short delays of 1.5 and 3.5 s) or 2.5 s (for long delays of 5.5 and 7.5 s) after the onset of the cue display. The orders of these sessions were counterbalanced across participants who completed both. Following the preparatory period, two superimposed gratings were then shown for 1 s. The target grating was shown with a small angular offset with respect to the cued reference orientation, whereas the distractor grating was shown in the uncued reference orientation (e.g., if rightward orientation was cued, the rightward grating was shown in 45° ± d and the leftward grating was shown in 135°). Note that the angular offset was determined individually based on the threshold obtained during the training session (at least 3 blocks, 30 trials/block), using a staircase procedure (Best Parameter Estimation by Sequential Testing, Best PEST), as implemented in the Palamedes Toolbox (*Prins and Kingdom, 2009*). Participants used a keypad to report whether the attended orientation was more leftward or rightward

relative to the reference orientation. Each trial was separated by an inter-trial interval of 3–7 s (2 s per step). Trial-by-trial feedback ('correct' or 'incorrect') was provided in the training session but not during scanning. Instead, the percentage of correct responses was provided at the end of each run in the scanning session to avoid the impact of trial-level feedback on neural activity.

Given the need to maximize the number of trials for fMRI-based MVPA, we could not accommodate additional conditions (e.g., neutral cue) to measure the behavioral effects of attention. However, our prior work using similar feature cueing paradigms (*Liu et al., 2007*; *Jigo et al., 2018*) found that attentional cueing improved behavioral performance relative to a neutral condition. Thus, it is highly likely that our well-trained participants used the cue to direct their attention in the fMRI experiment. Furthermore, our neural measures of attentional signals revealed feature-specific attentional modulations, further validating our approach. To prevent the cue-related sensory difference from contributing to neural activity, we reversed the mapping between colors and orientations halfway through the experiment (e.g., red indicated 'attend leftward orientation' and green indicated 'attend rightward orientation' in the first half of the runs, and vice versa for the second half of the runs), with the order counterbalanced across subjects. The mapping of colors and orientations was reversed only once in the middle of the experiment to prevent misremembering of the color-orientation associations. To reduce temporal expectancy over a fixed period, the preparatory period (i.e., cue-to-stimulus interval) varied from 1.5 to 7.5 s with different probabilities (10% for 1.5 or 3.5 s each, 40% for 5.5 or 7.5 s each). The long-delay trials (5.5 or 7.5 s) were selected for subsequent analyses, as they allow the separation of the preparatory activity from the grating-evoked response during fMRI scanning. The short-delay trials were included to encourage a sustained maintenance of attention throughout the entire preparation period.

## Perception task
On each trial of the perception task (*Figure 1B*), a single grating was shown for 1 s, followed by an inter-trial interval between 3 and 7 s. To equate the sensory inputs between attention and perception tasks, the orientation was shifted away from the reference orientation by the same angular offset as that used in the attention task with each individual participant's own threshold. Participants performed the same orientation discrimination task by comparing the single orientation to the reference orientation. We provided the percentage of correct responses at the end of each run as feedback.

## Eye tracking and analysis
To evaluate the stability of visual fixation, we used Eyelink Portable Duo system (SR Research, Ontario, Canada) to monitor each observer's eye position during the training session at a sampling rate of 500 Hz. One participant's data was not used due to the unstable recording of the eye. The data were then analyzed using custom Matlab code.

To examine whether participants adopted a space-based strategy during the preparatory period in the attention task, such as directing their gaze leftward in attend leftward trials, and vice versa for the attend rightward trials, we analyzed the participants' eye positions recorded during the training session. Horizontal and vertical eye positions were analyzed separately. Paired *t*-tests were performed to compare horizontal and vertical eye positions between two attention conditions. A two-way mixed ANOVA (2 sessions × 2 attended orientations) was applied to the horizontal and vertical positions, respectively.

## fMRI data acquisition
Imaging was performed on a Siemens 3T scanner (MAGNETOM Prisma, Siemens Healthcare, Erlangen, Germany) using a 20-channel coil at Zhejiang University (Hangzhou, China). For each participant, we acquired high-resolution T1-weighted anatomical images (field of view, 240 × 240 mm, 208 sagittal slices; 0.9 mm$^3$ resolution), T2*-weighted echo-planar functional images consisting of 46 slices (TR, 2 s; TE, 34 ms; flip angle, 50°; matrix size, 80 × 80; in-plane resolution, 3 × 3 mm; slice thickness, 3 mm, interleaved, no gap) and a 2D T1-weighted anatomical image (0.8 × 0.8 × 3 mm) for aligning functional data to high-resolution anatomical data.

## fMRI data preprocessing
Data analyses were performed using mrTools (*Gardner et al., 2018*) and custom code in Matlab. For each run, functional data were preprocessed with head motion correction, linear detrending, and

temporal high-pass filtering at 0.01 Hz. Data were converted to percentage signal change by dividing the time course of each voxel by its mean signals in each run. We concatenated the six runs of the attention task and the three runs of the perception task separately for further analysis. One of the attention runs in one subject was excluded due to low accurate performance (<50%).

## Definition of ROIs

### Visual and parietal ROIs

Following previous work (*Jigo et al., 2018*; *Gong and Liu, 2020b*; *Gong et al., 2022*), for each observer, we ran a separate retinotopic mapping session to obtain ROIs in occipital and parietal areas. Observers viewed four runs of rotating wedges (i.e., clockwise and counterclockwise) and two runs of rings (i.e., expanding and contracting) to map the polar angle and radial components, respectively (*DeYoe et al., 1996*; *Engel et al., 1997*; *Sereno et al., 1995*). Borders between areas were defined as the phase reversals in a polar angle map of the visual field. Phase maps were visualized on computationally flattened representations of the cortical surface, which were generated from the high-resolution anatomical image using FreeSurfer (http://surfer.nmr.mgh.harvard.edu) and custom Matlab code.

To help identify the topographic areas in parietal areas, we ran two runs of memory-guided saccade task modeled after previous studies (*Konen and Kastner, 2008*; *Schluppeck et al., 2006*; *Sereno et al., 2001*). Observers fixated at the screen center while a peripheral (~10° radius) target dot was flashed for 500 ms. The flashed target was quickly masked by a ring of 100 distractor dots randomly positioned within an annulus (8.5° – 10.5°). The mask remained on screen for 3 s, after which participants were instructed to make a saccade to the memorized target position, then immediately saccade back to the central fixation. The position of the peripheral target shifted around the annulus from trial to trial in either a clockwise or counterclockwise order. Data from the memory-guided saccade task were analyzed using the same phase encoding method as the wedge and ring data. Therefore, the following ROIs in each hemisphere were identified after the completion of this session: V1, V2, V3, V3A/B, V4, V7/IPS0, and IPS1–IPS4.

### Frontal ROIs

Following previous work (*Jigo et al., 2018*; *Gong and Liu, 2020b*; *Gong et al., 2022*), we used a deconvolution approach by fitting each voxel's time series from the attention task with a general linear model (GLM) to determine the event-related activations in the brain (see Supplementary materials: Deconvolution). For each voxel, we computed the goodness of fit measure ($r^2$ value), which indicates the amount of variance explained by the deconvolution model (*Gardner et al., 2018*). The $r^2$ value represents the degree to which the voxel's time series is correlated with the task events, regardless of any differential responses among conditions. Based on the task-related activation (as indexed by $r^2$ value) and anatomical criteria, we defined two frontal areas in each hemisphere that were active during the attention task: one is located superior to the precentral sulcus and near the superior frontal sulcus (FEF) and the other is located toward the inferior precentral sulcus, close to the junction with the inferior frontal sulcus (IFJ).

### Groups of region

To characterize the patterns of neural response across cortical hierarchy and streamline data presentation, we grouped results from the nine areas into four groups based on functional and anatomical considerations: primary visual cortex (V1); EVC, consisting of V2, V3, V3ab, and V4; IPS, consisting of IPS0–IPS4; PFC, consisting of FEF and IFJ. Individual areas within each group exhibited qualitatively similar results.

Note that we analyzed V1 separately for two reasons. First, previous studies consistently identify V1 as the main locus of sensory-like templates during feature-specific preparatory attention (*Kok et al., 2014*; *Aitken et al., 2020*). Second, V1 shows the strongest orientation selectivity within the visual hierarchy (*Priebe, 2016*). In contrast, the EVC (comprising V2, V2, V3AB, and V4) demonstrates broader selectivity for complex features (*Grill-Spector and Malach, 2004*). Therefore, it would be particularly informative to analyze V1 separately for our orientation-based attention paradigm.

## Multivoxel pattern analysis

### Decoding of attended orientation

To test if multivariate patterns of activity represent information of the attended orientation, separate MVPA analyses were applied on the activity patterns for the preparation and stimulus selection periods. Following previous work (*Jigo et al., 2018*; *Gong and Liu, 2020b*; *Gong et al., 2022*), for this analysis, we extracted fMRI signals from raw time series in the long delay trials with correct behavioral responses (~72 trials per attention condition); short-delay trials were excluded as they could not provide enough data points to measure preparatory activity. We then obtained averaged BOLD response in a 2-s window for each voxel and each trial in a given ROI, separately for preparatory activity (4–6 s after the onset of the cue) and stimulus-evoked activity (4–6 s after the onset of the gratings). The response amplitudes across two attention conditions in each ROI were further z-normalized, separately for the preparation and stimulus-related activity. These normalized single-trial BOLD responses were used for the MVPA. We trained a classifier using the Fisher linear discriminant (FLD) analysis to discriminate between two attended orientations (leftward vs. rightward) and tested its performance with a leave-one-run-out cross-validation scheme. This process was repeated until each run was tested once and the decoding accuracy (i.e., the proportion of correctly classified trials) was averaged across the cross-validation folds. The statistical significance of decoding accuracy was evaluated by comparing it to the chance level obtained from a permutation test (see Permutation test). To assess if the decoding accuracy differed between No-Ping and Ping experiments, we performed two-way mixed ANOVAs (2 sessions × 4 regions) on the decoding accuracy.

### Cross-task generalization from the perception task to attention task

Following previous work (*Jigo et al., 2018*; *Gong et al., 2022*), to test whether the neural patterns in the preparatory and stimulus selection periods from the attention task reflected sensory processing of isolated features, we trained an FLD classifier using the normalized BOLD responses from the perception task (4–6 s after the trial onset) to discriminate leftward versus rightward orientation. Then, we tested this classifier on the normalized response from the independent runs of the attention task to discriminate between attend leftward versus attend rightward orientations, separately for preparation and stimulus selection periods. The significance of decoding accuracy was compared to the chance level obtained from a permutation test (see Permutation test). To assess if the generalization performance differed between No-Ping and Ping sessions, we performed two-way mixed ANOVAs (2 sessions × 4 regions) on the decoding accuracy.

### Neural distance between attended and perceived orientations

The decoding accuracy from the cross-task generalization test reflects a discretized readout of the pattern similarity between different conditions. However, employing continuous similarity measures, such as Mahalanobis distance (*Mahalanobis, 1936*), could be more reliable compared to decoding accuracy (*Walther et al., 2016*). Therefore, we calculated the Mahalanobis distance to quantify the pattern similarity between two attended orientations and two perceived orientations. For each participant and each ROI, we have M points (i.e., M trials for each attended orientation) in the $N$-dimensional space ($N$ = 100, number of voxels). For each data point in the attended orientation condition, we computed its distance to each of the orientation distributions (from the perception task). Averaged distance values were then calculated for each combination of attended orientation and perceived orientation pairs. A sensory-like hypothesis would predict smaller distance between the distribution of the attended orientation (e.g., attend leftward) and the distribution of corresponding orientation (e.g., perceive leftward) compared to the alternative orientation (e.g., perceive rightward). Two-way repeated-measures ANOVA (2 attended orientations × 2 perceived orientations) was applied on the Mahalanobis distance, separately for each region and each session.

### Neural–behavioral relationships

We tested if the representation format during preparatory attention was associated with subsequent behavior. For each trial, we calculated the Mahalanobis distance between the attention conditions (attend leftward and attend rightward) and the perceived orientations (leftward and rightward orientation). We estimated the AMI based on these distance values. This index measures how much attention

modulated the pattern similarity for the Same orientation condition (e.g., attend and perceive the same orientation) relative to the Different orientation condition (e.g., attend and perceive different orientations). The index was calculated as follows: AMI = $(D_{different} - D_{same})/(D_{different} + D_{same})$, where $D_{same}$ and $D_{different}$ are the measured distance ($D$) in the Same and Different orientation condition, respectively. Next, we tested the behavioral relevance of AMI in two ways: (1) Inter-subject analysis: correlated AMI with RT and accuracy across participants, separately for each session; (2) Within-subject analysis: for each participant, we sorted the single-trial AMI values in descending order and selected top-ranked 25% trials and bottom-ranked 25% trials to represent 'strong modulation' and 'weak modulation' trials, respectively. We then extracted behavioral responses on these selected trials and calculated RT and accuracy for each trial type. Paired $t$-tests were used to compare between 'strong modulation' and 'weak modulation' trials in each session.

## IC analysis

We used IC to examine shared changes in pattern discriminability over time, a method that allows inference based on multivoxel pattern information rather than overall BOLD response (*Jia et al., 2020*; *Ng et al., 2021*). To track the flow of multivariate information across time (i.e., across trials), we measured the fluctuations (covariance) in pattern-based discriminability by calculating the Mahalanobis distance of each trial to the two attended orientations, using a leave-one-run-out cross-validation scheme. For each ROI, we calculated the Mahalanobis distance between the pattern of activity for each attention trial from one left-out run and the distribution of each attended orientation of the remaining runs. To quantify the degree of attentional modulation, we calculated the AMI using the same formula as mentioned above, where $D_{same}$ and $D_{different}$ are the measured distance ($D$) in the Same and Different condition. This index measures how much the pattern similarity increased for the same attention condition (e.g., attend leftward to attend leftward) relative to the different attention condition (e.g., attend leftward to attend rightward). A positive AMI indicates relative proximity to the same attention condition, whereas a negative AMI indicates relative proximity to the different attention condition. A time course of AMI values was generated across runs and pairwise correlated between ROIs using Pearson correlation analysis and Fisher $z$-transformed. Independent $t$-tests were used to compare the connectivity between No-Ping and Ping sessions. To assess the relationship between ICs and behavior, we correlated ICs with RT and accuracy across participants, separately for each session.

## Permutation test to evaluate classifier performance

Following previous work (*Jigo et al., 2018*; *Gong et al., 2022*), for each brain area, we evaluated the statistical significance of the observed decoding accuracy using a permutation test scheme. We first shuffled the trial labels in the training data and trained the same FLD classifier on the shuffled data. We then tested the classifier on the (unshuffled) test data to obtain decoding accuracy. For each ROI and each participant, we repeated this procedure 1000 times to compute a null distribution of decoding accuracy. To compute the group-level significance, we averaged the 20 null distributions to obtain a single null distribution of 1000 values for each ROI. To determine if the observed decoding accuracy significantly exceeds the chance level, we compared the observed value to the 95 percentiles of this group-level distribution (corresponding to p = 0.05). Note that these ROIs were pre-defined with strong priors as their activation in attention tasks has been consistently reported in the literature. Nevertheless, for those analyses where multiple comparisons were performed across regions, we applied a Bonferroni correction to adjust the p-values.

## Bayesian analysis

To evaluate the strength of evidence for the null hypothesis, we conducted Bayesian analyses (*Wagenmakers, 2007*) using standard priors as implemented in JASP Version 0.17.1 (*JASP Team, 2023*). We performed Bayesian $t$-tests and computed Bayes factor ($BF_{01}$) to compare between two attention conditions (attend leftward vs. attend rightward). Additionally, we used Bayesian repeated-measures ANOVA and computed the exclusion Bayes factors ($BF_{excl}$) to assess the evidence for excluding specific effects across all models. A Bayes factor (BF) greater than 1 provides support for the null hypothesis. Specifically, a BF between 1 and 3 indicates weak evidence, a BF between 3 and 10 indicates moderate evidence, and a BF greater than 10 indicates strong evidence (*van Doorn et al., 2021*).

## Approach to handle partially overlapped samples

Our study used partially overlapping samples, with 14 out of 20 participants completing both No-Ping and Ping sessions, while the remainder completed one of the two sessions. The most important analyses entailed assessing whether decoding accuracy was above chance, for which we used the permutation-based method (see above) within each session. Thus, these analyses were unaffected by the partially overlapping samples. In a few analyses where we compared across sessions, we used statistical tests treating 'session' as a between-subject factor. We believe this is a reasonable approach, as a between-subject test is more conservative than a within-subject test, such that any significant effect emerged should be a genuine effect. To be certain, we also conducted additional analyses with 'session' as a within-subject factor on the subset of data from the 14 participants who completed both sessions in a counterbalanced order. The results were highly similar to those reported in the main text.

## Acknowledgements

This work was supported by National Science and Technology Innovation 2030—Major Project 2021ZD0200409, National Natural Science Foundation of China (32371087, 32300855, 3200784), Fundamental Research Funds for the Central University (226-2024-00118), a grant from the MOE Frontiers Science Center for Brain Science & Brain-Machine Integration at Zhejiang University and Non-profit Central Research Institute Fund of Chinese Academy of Medical Sciences 2023-PT310-01.

## Additional information

### Funding

| Funder | Grant reference number | Author |
|---|---|---|
| National Science and Technology Innovation 2030 | Major Project 2021ZD0200409 | Mengyuan Gong |
| National Natural Science Foundation of China | 32371087 | Mengyuan Gong |
| National Natural Science Foundation of China | 32300855 | Ke Jia |
| National Natural Science Foundation of China | 3200784 | Mengyuan Gong |
| Fundamental Research Funds for the Central University | 226-2024-00118 | Mengyuan Gong |
| MOE Frontiers Science Center for Brain Science & Brain-Machine Integration | | Mengyuan Gong |
| Chinese Academy of Medical Sciences | Non-profit Central Research Institute Fund 2023-PT310-01 | Ke Jia |

The funders had no role in study design, data collection, and interpretation, or the decision to submit the work for publication.

### Author contributions

Yilin Chen, Data curation, Formal analysis, Investigation; Taosheng Liu, Writing – original draft, Writing – review and editing; Ke Jia, Conceptualization, Funding acquisition, Writing – review and editing; Jan Theeuwes, Writing – review and editing; Mengyuan Gong, Conceptualization, Funding acquisition, Writing – original draft, Project administration, Writing – review and editing

### Author ORCIDs

Yilin Chen ⓘ https://orcid.org/0000-0002-5695-7093
Ke Jia ⓘ https://orcid.org/0009-0005-5498-8952

Mengyuan Gong https://orcid.org/0000-0003-0333-4957

### Ethics

Participants provided written informed consent according to the study protocol approved by the Institutional Review Board at Zhejiang University (2020-06-001).

Reviewer #1 (Public review): https://doi.org/10.7554/eLife.103425.4.sa1
Reviewer #3 (Public review): https://doi.org/10.7554/eLife.103425.4.sa2
Author response https://doi.org/10.7554/eLife.103425.4.sa3

## Additional files

### Supplementary files

MDAR checklist

### Data availability

All data, analyses, and task codes have been made publicly available via the Open Science Framework at https://osf.io/ghaxv/.

The following dataset was generated:

| Author(s) | Year | Dataset title | Dataset URL | Database and Identifier |
|---|---|---|---|---|
| Gong M | 2025 | Data from dual-format attentional template during preparation | https://doi.org/10.17605/OSF.IO/RDQFS | Open Science Framework, 10.17605/OSF.IO/RDQFS |

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

# Appendix 1

## Univariate analysis

### Deconvolution

For the data in the attention task, we used a deconvolution approach by fitting each voxel's time series with a general linear model (GLM) with seven regressors, four corresponding to the long delay (5.5 and 7.5 s) trials with correct responses (attended orientation × delay), two corresponding to the short delay (1.5 and 3.5 s) trials with correct responses, and the remaining four regressors corresponding to incorrect trials for each length of the delay. Each trial was modeled by a set of 12 finite impulse response functions (FIR) after the trial onset. For the perception task data, we used a GLM with two regressors, corresponding to two features (leftward vs. rightward). Each trial was modeled by a set of eight FIR after the trial onset. The design matrix was pseudo-inversed and multiplied by the time series to obtain an estimate of the hemodynamic response evoked by each condition. Correct trials from the attention task and all trials from the perception task were entered into further univariate and multivariate analysis.

### Attentional modulations on BOLD response

To select active voxels during the task, we discarded noisy voxels with mean responses larger than 10% signal change. Because the stimuli were presented at the center of the display, areas across two hemispheres were combined. Using the $r^2$ values from the deconvolution analysis to index the relevance of each voxel to the task, voxels in each brain area were sorted in descending order of their $r^2$ values. The top-ranked 100 voxels in each combined ROI were then selected for further analyses. The results remained highly similar when using different numbers of voxels (e.g., 80 or 120).

To assess whether feature-based attention modulated overall BOLD response during the preparatory and stimulus selection periods, we averaged the fMRI response across voxels for long-delay trials of the attention task, separately for each ROI, each attention condition, and each time point. Then, we averaged a time window of 4–6 s after the onset of the cue to index the preparatory activity, and averaged a time window of 4–6 s after the onset of superimposed gratings to index the stimulus-evoked response. To test the influence of attention in different brain regions, we applied two-way repeated-measures ANOVAs (attended orientation × region) on BOLD response, separately for activity during preparatory and stimulus selection periods.

## Univariate results

During the No-Ping session, when examining the group-level fMRI time courses, we observed slightly elevated activity during the preparation period, particularly in higher-order frontoparietal areas, followed by a robust response to the superimposed gratings (*Appendix 1—figure 1*). To examine whether feature-based attention modulated overall BOLD in the attention task across regions, we conducted two-way repeated-measures ANOVAs (attended orientation × region) on the BOLD response during preparation and stimulus periods, separately. The results showed a main effect of region in the preparation period ($F(3,57) = 6.748$; p = 0.001; $\eta_p^2 = 0.262$) and stimulus periods ($F(3,57) = 38.230$; p < 0.001; $\eta_p^2 = 0.668$). No other effects were significant in both periods (preparation: ps > 0.526; stimulus: ps > 0.135). These results demonstrate stronger response in higher-order frontoparietal areas than in sensory areas. The lack of attention modulation on BOLD response suggests that attending to specific features did not influence mean neural activity during the preparation and stimulus selection periods. These results are consistent with our previous studies using similar paradigms.

The presence of visual impulse during preparation in the Ping session yielded a similar time course of neural activity (*Appendix 1—figure 1*). Consistent with the observation in No-Ping session, two-way repeated-measures ANOVAs revealed a main effect of region in both the preparation ($F(3,57)$ = 3.445; p = 0.023; $\eta_p^2 = 0.153$) and stimulus periods ($F(3,57) = 29.911$; p < 0.001; $\eta_p^2 = 0.612$). No other effects were significant in both periods (preparation: ps > 0.462; stimulus: ps > 0.063). These results suggest that a briefly presented impulse stimulus caused little changes to the neural response amplitude and promoted us to further investigate multivariate neural activity patterns.

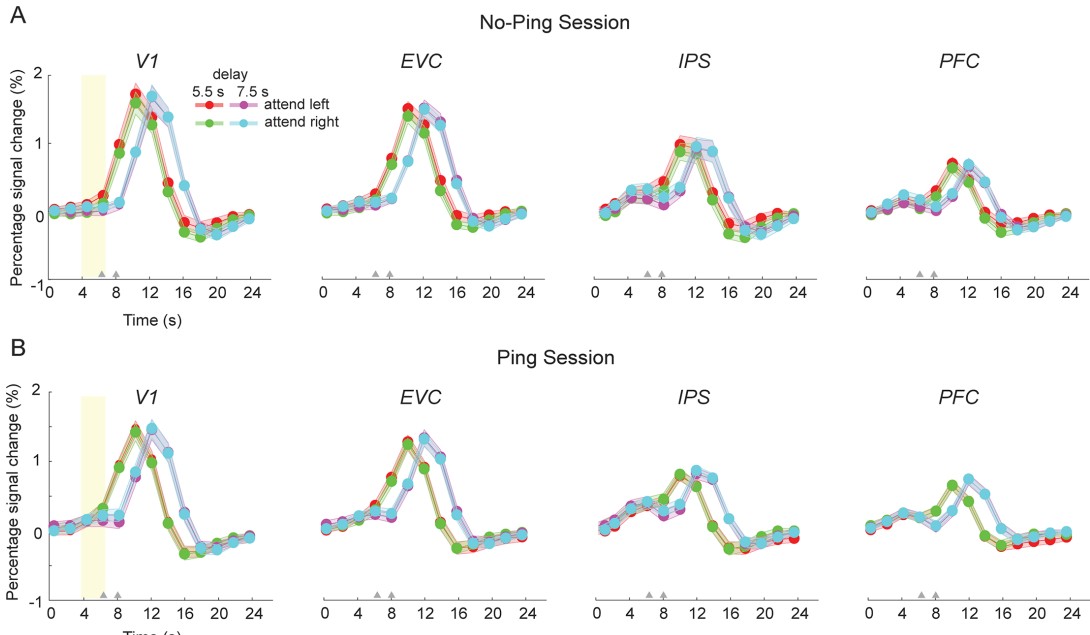

**Appendix 1—figure 1.** Univariate BOLD results. (**A**). Mean fMRI time course from four regions (V1, EVC, IPS, and PFC) for No-Ping session and (**B**) Ping session. The gray triangle indicates the onset of superimposed gratings. The gray area indicates a slightly elevated response in the presence of visual impulse. Error bars denote standard error of the means.

## Analysis of eye movement

Although participants were required to maintain stable fixation in our experiment, they might have shifted their spatial attention differently when preparing to attend different features (e.g., eyes moved toward leftward or rightward). To assess the potential influence of overt spatial attention in our results, we analyzed the participants' eye position recorded during the training session and observed no significant difference between attended features during preparation in the No-Ping (p = 0.328, $BF_{01}$ = 2.760 for horizontal; p = 0.231, $BF_{01}$ = 2.210 for vertical; *Appendix 1—figure 2*) and Ping sessions (p = 0.450, $BF_{01}$ = 3.229 for horizontal; p = 0.588, $BF_{01}$ = 3.675 for vertical). To examine whether there were any differences in eye movement between two sessions, a two-way mixed ANOVA (sessions × attended orientation) was applied to the eye position data. The analyses revealed neither main effects nor an interaction effect for the horizontal (ps > 0.222, $BF_{excl}$ > 1.593) and vertical eye movements (ps > 0.158, $BF_{excl}$ > 1.185). We note that we only collected eye-tracking data during behavioral training and not during scanning due to technical limitations. However, it seems unlikely that the participants would adopt a different eye fixation strategy after training, especially given the centrally presented stimulus configuration that made eye movement an ineffective strategy.

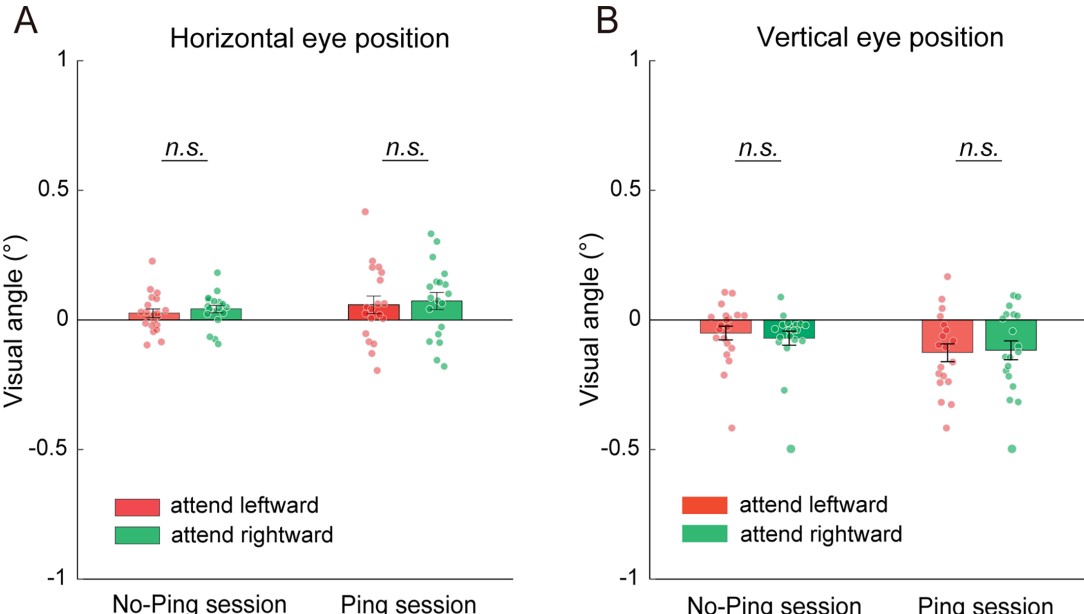

**Appendix 1—figure 2.** Analysis of eye positions. (**A**) Group-level horizontal eye position and (**B**) vertical eye position for the preparation period during behavioral training in No-Ping and Ping sessions. Each dot represents one participant. Error bars denote standard error of the means.

