## [Editor Report · eLife Assessment]

By combining the 'pinging' technique with fMRI-based multivariate pattern analysis, this **important** study provides **compelling** evidence for a dual-format representation of attention during the preparatory period. The findings help reconcile the debate between sensory-like and non-sensory accounts of attentional templates and shed light on how the brain flexibly deploys different forms of templates to guide attention. This work will be of broad interest to researchers in psychology, vision science, and cognitive neuroscience.

---

## [Referee Report · Reviewer #1 (Public review)]

Summary:

The aim of the experiment reported in this paper is to examine the nature of the representation of a template of an upcoming target. To this end, participants were presented with compound gratings (consisting of tilted to the right and tilted to the left lines) and were cued to a particular orientation - red left tilt or blue right tilt (counterbalanced across participants). There two directly compared conditions: (i) no ping: where there was a cue, that was followed by a 5.5-7.5s delay, then followed by a target grating in which the cued orientation deviated from the standard 45 degrees; and (ii) ping condition in which all aspects were the same with the only difference that a ping (visual impulse presented for 100ms) was presented after the 2.5 seconds following the cue. There was also a perception task in which only the 45 degrees to the right or to the left lines were presented. It was observed that during the delay, only in the ping condition, were the authors able to decode orientation of the to be reported target using the cross-task generalization. Attention decoding, on the other hand, was decoded in both ping and non-ping conditions. It is concluded that the visual system has two different functional states associated with a template during preparation: a predominantly non-sensory representation for guidance and a latent sensory-like for prospective stimulus processing.

Strengths:

There is so much to be impressed with in this report. The writing of the manuscript is incredibly clear. The experimental design is clever and innovative. The analysis is sophisticated and also innovative -the cross-task decoding, the use of Mahalanobis distance as a function of representational similarity, the fact that the question is theoretically interesting, the excellent figures.

Comments on revisions:

I have no further comments.

---

## [Referee Report · Reviewer #3 (Public review)]

This paper discusses how non-sensory and latent, sensory-like attentional templates are represented during attentional preparation. Using multivariate pattern analysis, they found that visual impulses can enhance the decoding generalization from perception to attention tasks in the preparatory stage in the visual cortex. Furthermore, the emergence of the sensory-like template coincided with enhanced information connectivity between V1 and frontoparietal areas and was associated with improved behavioral performance. It is an interesting paper with supporting evidence for the latent, sensory-like attentional template.

Comments on revisions:

I appreciate the authors' thoughtful revisions, which have addressed my earlier concerns. I have no further comments.

---

## [Author Response]

The following is the authors’ response to the previous reviews

**Reviewer #1 (Public review):**
I am impressed with the thoroughness with which the authors addressed my concerns. I don't have any further concerns and think that this paper makes an interesting and significant contribution to our understanding of VWM. I would only suggest adding citations to the newly added paragraph where the authors state "It could be argued that preparatory attention relies on the same mechanisms as working memory maintenance." They could cite work by Bettencourt and Xu, 2016; and Sheremata, Somers, and Shomstein (2018).

We thank the reviewer for the positive feedback. We have now cited the referenced work in the manuscript (Page. 19, Line 371).

**Reviewer #2 (Public review):**
Overall, I think that the authors' revision has addressed most, if not all, of my major concerns noted in my previous comments. The results appear convincing and I do not have additional comments.

We thank the reviewer for the positive feedback and are pleased that the revision addressed the major concerns.

**Reviewer #3 (Public review):**
(1) The authors addressed most of my previous concerns and provided additional data analysis. They conducted further analyses to demonstrate that the observed changes in network communication are associated with behavioral RTs, supporting the idea that the impulse-driven sensory-like template enhances informational connectivity between sensory and frontoparietal areas, and relates to behavior.

We are pleased that the revision addressed the major concerns.

(2) I would like to further clarify my previous points regarding the definition of the two types of templates and the evidence for their coexistence. The authors stated that the sensory-like template likely existed in a latent state and was reactivated by visual pings, proposing that sensory and non-sensory templates coexist. However, it remains unclear whether this reflects a dynamic switch between formats or true coexistence. If the templates are non-sensory in nature, what exactly do they represent? Are they meant to be abstract or conceptual representations, or, put simply, just "top-down attentional information"? If so, why did the generalization analysestraining classifiers on activity during the stimulus selection period and testing on preparatory activity-fail to yield significant results? While the stimulus selection period necessarily encodes both target and distractor information, it should still contain attentional information. I would appreciate more discussion from this perspective.

We thank the reviewer for the helpful clarification of previous comments. Since we addressed similar comments from Reviewer 2 (Point 2) in the previous round, our response below may appear somewhat repetitive. First, regarding whether our findings reflect a dynamic switch between non-sensory and sensory-like template, or the ‘coexistence’ of two template formats, we acknowledge that the temporal limitations of fMRI prevent us from directly testing dynamic representations. However, several aspects of our data favor the latter interpretation: (1) our key findings remained consistent in the subset of participants (N=14) who completed both No-Ping and Ping sessions in counterbalanced order. This makes it unlikely that participants systematically switched cognitive strategies (e.g., using non-sensory templates in the No-Ping session versus sensory-like templates in the Ping session) in response to the taskirrelevant, uninformative visual impulse; (2) while we agree that the temporal dynamics between the two templates remain unclear, it is difficult to imagine that orientation-specific templates observed in the Ping session emerged de novo from purely non-sensory templates and an exogenous ping. In other words, if there is no orientation information at all to begin with, how does it come into being from an orientation-less external ping? A more parsimonious explanation is that orientation information was already present in a latent format and was activated by the ping, in line with the models of “activity-silent” working memory. However, since the detailed circuit-level mechanism underlying such reactivation remain unclear, we acknowledge that this interpretation warrants direct investigation in future studies. This point is discussed in the main texts (Page 19-20, Line 389-402).

Second, while our data cannot definitively determine the nature of the non-sensory template, we consider categorical coding a plausible candidate based on prior visual search studies. For instance, categorical attributes (e.g., left-tilted vs. right-tilted) have been shown to effectively guide attention in orientation search tasks (Wolfe et al., 1992), similar to our paradigm. Further, categorical templates are more tolerant of stimulus variability, making them well-suited to our task, which involved trial-by-trial variations in target orientation around a reference (see Page 21, Line 427- 437 for more detailed discussions).

Third, the lack of generalization from stimulus selection to preparatory attention in the Ping session may relate to the limited overlap in shared information between these two periods. Neural activity during stimulus selection encodes sensory information about both orientations, along with sensory-like attentional signals (as indicated by the attention decoding and crosstask generalization from perception task to the stimulus-selection period). In contrast, preparatory activity likely involves a dominant non-sensory template, a latent sensory-like template, and residual sensory effects from the impulse stimulus. The limited overlap in sensory-like attentional signals may therefore be insufficient to support generalization across the two periods.

**Reviewer #2 (Recommendations for the authors)**
I think the central prediction of greater pattern similarity between 'attend leftward' and 'perceived leftward' in the ping session in comparison to the no-ping session the same also holds for 'attend rightward' and 'perceived rightward' could be directly examined by a two-way ANOVA (session × the attend orientation is the same/different from the perceived orientation) for each ROI (V1 and EVC). A three-way ANOVA might complicate readers' intuitive understanding of the implications of the statistical results.

We thank the reviewer for the suggestion. Following the reviewer’s suggestion, we defined a new condition label based on orientation consistency between attended and perceived orientations: (1) same orientation: averaging “attend leftward/perceive leftward” and “attend rightward/perceive rightward”; and (2) different orientation: averaging “attend leftward/perceive rightward” and “attend rightward/perceive leftward”. A two-way mixed ANOVA (session × orientation consistency) on Mahalanobis distance revealed a main effect of orientation consistency in V1 (F(1,38) = 4.21, p = 0.047, η_p_^2^ = 0.100), indicating that activity patterns were more similar when attended and perceived orientations matched. No significant main effect of session was found (p = 0.923). Importantly, a significant interaction was found in V1 (F(1,38) = 5.00, p = 0.031, η_p_^2^ = 0.116), suggesting that visual impulse enhanced the similarity between preparatory attentional template and the perception of corresponding orientation. In EVC, the same analysis revealed only a main effect of orientation consistency (F(1,38) = 5.87, p = 0.020, η_p_^2^ = 0.134), with no significant other effects (ps > 0.240). The interaction results were consistent with those reported in the original three-way ANOVA. We have now replaced the previous analysis with the new one in the main texts (Page 11-12, Line 231-242).